# Apelin signaling dependent endocardial protrusions promote cardiac trabeculation in zebrafish

**Jialing Qi, Annegret Rittershaus[†], Rashmi Priya[‡], Shivani Mansingh[§], Didier YR Stainier\*, Christian SM Helker\***

Department of Developmental Genetics, Max Planck Institute for Heart and Lung Research, Bad Nauheim, Germany

**\*For correspondence:**
Didier.Stainier@mpi-bn.mpg.de
(DYRS);
christian.helker@biologie.uni-marburg.de (CSMH)

**Present address:** [†]Philipps-University Marburg, Faculty of Biology, Cell Signaling and Dynamics, Marburg, Germany; [‡]The Francis Crick Institute, Organ Morphodynamics Laboratory, London, United Kingdom; [§]Biozentrum, University of Basel, Basel, Switzerland

**Abstract** During cardiac development, endocardial cells (EdCs) produce growth factors to promote myocardial morphogenesis and growth. In particular, EdCs produce neuregulin which is required for ventricular cardiomyocytes (CMs) to seed the multicellular ridges known as trabeculae. Defects in neuregulin signaling, or in endocardial sprouting toward CMs, cause hypotrabeculation. However, the mechanisms underlying endocardial sprouting remain largely unknown. Here, we first show by live imaging in zebrafish embryos that EdCs interact with CMs via dynamic membrane protrusions. After touching CMs, these protrusions remain in close contact with their target despite the vigorous cardiac contractions. Loss of the CM-derived peptide Apelin, or of the Apelin receptor, which is expressed in EdCs, leads to reduced endocardial sprouting and hypotrabeculation. Mechanistically, neuregulin signaling requires endocardial protrusions to induce extracellular signal-regulated kinase (Erk) activity in CMs and trigger their delamination. Altogether, these data show that Apelin signaling-dependent endocardial protrusions modulate CM behavior during trabeculation.

## Editor's evaluation

This first formal dissection of endocardial protrusions in zebrafish hearts describes how they anchor to cardiomyocytes, and how they participate in signaling pathways involved in trabeculation. The work combines elegant zebrafish reporters and high-quality imaging, as well as mutant lines and pathway inhibitors to provide key findings of how mutual regulation between the myocardium and the endocardium contribute to understanding of mechanisms underlying organ development. This manuscript is of broad interest to readers who study cardiogenesis and developmental biology.

## Introduction

To meet the needs of the growing embryo, the vertebrate heart has to undergo a series of complex morphogenetic events to transform from a linear tube into a mature organ. During cardiac trabeculation, CMs in the outer curvature of the ventricles delaminate toward the lumen to form multicellular sponge-like projections, called trabeculae (*Sedmera and Thomas, 1996*; *Sedmera et al., 2000*; *Stankunas et al., 2008*; *Liu et al., 2010*; *Peshkovsky et al., 2011*; *Staudt et al., 2014*). Cardiac trabeculae are crucial to achieve increased contractility as well as for the formation of the conduction system. Trabeculation defects are often associated with left ventricular noncompaction (*Oechslin et al., 2000*; *Stöllberger and Finsterer, 2004*), embryonic heart failure, and lethality (*Gassmann et al., 1995*; *Lee et al., 1995*; *Lai et al., 2010*; *Liu et al., 2010*; *Rasouli and Stainier, 2017*).

In zebrafish, as in other vertebrates, the early embryonic heart consists of two cell layers, the myocardium and the endocardium, separated by a specialized extracellular matrix called the cardiac jelly (CJ) (*Stainier and Fishman, 1992*; *Brutsaert et al., 1996*). Recently, it has been shown that endocardial cells (EdCs), similar to blood endothelial cells (ECs), form sprouts, and that these sprouts are mostly oriented toward the myocardium (*Del Monte-Nieto et al., 2018*). During sprouting angiogenesis, ECs first extend filopodia to sense the microenvironment for growth factors, then they migrate into avascular areas and form new blood vessels (*Gerhardt et al., 2003*). Due to its similarity to sprouting angiogenesis, the sprouting of EdCs has been termed endocardial sprouting. However, whether endocardial sprouting is regulated by the same signaling pathways as sprouting angiogenesis is not known.

Multiple signaling pathways have been implicated in cardiac trabeculation, including neuregulin (Nrg)/ErbB signaling. Mouse and zebrafish embryos lacking the endocardium-derived ligand Nrg or the ErbB receptor, which is expressed by the myocardium, fail to form trabeculae (*Gassmann et al., 1995*; *Lee et al., 1995*; *Meyer and Birchmeier, 1995*; *Lai et al., 2010*; *Liu et al., 2010*; *Rasouli and Stainier, 2017*). Furthermore, endocardial Notch signaling (*Grego-Bessa et al., 2007*; *D'Amato et al., 2016*; *Del Monte-Nieto et al., 2018*), angiopoietin 1/Tie2 signaling (*Suri et al., 1996*; *Tachibana et al., 2005*; *Qu et al., 2019*), and semaphorin 3E/plexinD1 signaling (*Sandireddy et al., 2019*) are required for cardiac trabeculation in mouse. Of note, genetic deletion of the relevant receptors in the endocardium results in attenuated endocardial sprouting (*Qu et al., 2019*) and trabeculation defects (*Grego-Bessa et al., 2007*; *D'Amato et al., 2016*; *Del Monte-Nieto et al., 2018*; *Qu et al., 2019*; *Sandireddy et al., 2019*).

Cells communicate by a variety of mechanisms including paracrine and contact-dependent signaling. More recently, a novel mechanism of cell communication by active transport of signaling molecules through filopodia-like actin-rich membrane protrusions, also known as cytonemes, has been shown in different models including *Drosophila* (*Ramírez-Weber and Kornberg, 1999*; *Roy et al., 2011*; *Huang et al., 2019*), chick (*Sanders et al., 2013*), zebrafish (*Stanganello et al., 2015*), and mouse (*Fierro-González et al., 2013*). Like filopodia, cytonemes depend on actin polymerization by various effector proteins including formins, profilin, and IRSp53, a substrate for the insulin receptor (*Rottner et al., 2017*).

In this study, we take advantage of the zebrafish model, as its transparency allows single-cell resolution and high-speed imaging of the beating heart, to analyze endocardial-myocardial communication during embryogenesis. By investigating *apelin* (*apln*) mutants, we found that endocardial protrusion formation is controlled by Apln signaling. We also observed by in vivo imaging that endocardial protrusions promote cardiac trabeculation by modulating Nrg/ErbB/Erk signaling. Altogether, our results provide new insights into the role of endocardial protrusions during cardiac trabeculation.

## Results

### Endocardial-myocardial interactions in zebrafish

The early embryonic heart in vertebrates is composed of two cell types: EdCs and myocardial cells (*Figure 1A–D*). In order to analyze possible interactions between the endocardial and myocardial cells in zebrafish, we genetically labeled the actin cytoskeleton of the EdCs using the *TgBAC(cdh5:Gal4ff)* and *Tg(UAS:LIFEACT-GFP)* lines, and the membrane of myocardial cells with mCherry using the *Tg(myl7:mCherry-CAAX)* line. We observed endocardial protrusions extending toward the myocardium at 24 (*Figure 1A–A"*) and 48 (*Figure 1B–B"*, *Figure 1—figure supplement 1A*) hours post-fertilization (hpf). Of note, we observed more endocardial protrusions in the ventricle than in the atrium at 48, 60, and 72 hpf (*Figure 1—figure supplement 1B*). Subsequently, these ventricular endocardial protrusions formed anchor points with the myocardium, and to be consistent with similar observations in mouse (*Del Monte-Nieto et al., 2018*) we will refer to them as touchdowns (*Figure 1B–B"*). Notably, these touchdowns are stable even during cardiac contractions (*Figure 1E–H*, *Figure 1—video 1*). Starting at around 60 hpf, cardiomyocytes (CMs) delaminate from the compact layer toward the lumen to seed the trabecular layer (*Figure 1C, C' and C""*), as reported before (*Liu et al., 2010*; *Peshkovsky et al., 2011*; *Staudt et al., 2014*; *Priya et al., 2020*). At this stage, we observed that endocardial protrusions appear to extend along, and sometimes around, the delaminating CMs (*Figure 1C" and C""*, *Figure 1—video 2*). Next, trabecular CMs start to assemble into

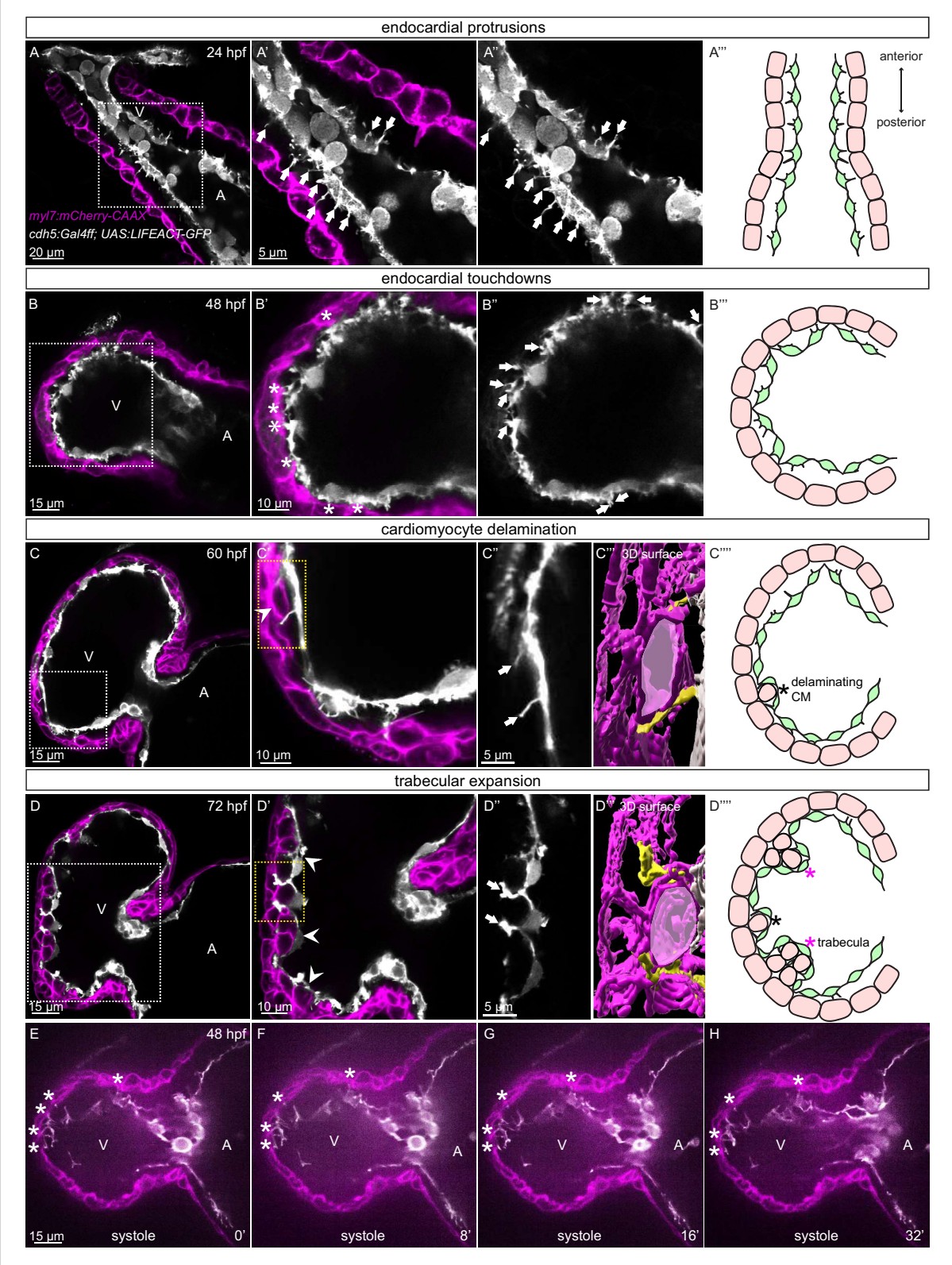

**Figure 1.** Stages of endocardial-myocardial interactions during zebrafish heart development. (**A-D**) Confocal projection images of the heart of Tg(myl7:mCherry-CAAX); Tg(cdh5:Gal4ff); Tg(UAS:LIFEACT-GFP) zebrafish at 24 (**A**), 48 (**B**), 60 (**C**) and 72 (**D**) hpf. (**A-A''**) Endocardial protrusions (arrows) towards the myocardium at 24 hpf. (**B-B''**) Endocardial protrusions (arrows) and touchdowns (asterisks) with the myocardium at 48 hpf. (**C-C'''**) Endocardial protrusions (arrows) during CM delamination (arrowheads) at 60 hpf. (**C'''**) 3D surface rendering of the area in the yellow box in **C'**. (**D-D'''**)

*Figure 1 continued on next page*

*Figure 1 continued*

Endocardial protrusions (arrows) during trabecular assembly and expansion (arrowheads) at 72 hpf. (**D'''**) 3D surface rendering of the area in the yellow box in **D'**. (**A''''-D''''**) Schematics of endocardial protrusions, endocardial touchdowns, CM delamination, and trabecular expansion. Black asterisks indicate delaminating CMs; purple asterisks indicate trabeculae. (**E-H**) Still images from a spinning disc time-lapse movie of a 48 hpf Tg(myl7:mCherry-CAAX); Tg(cdh5:Gal4ff); Tg(UAS:LIFEACT-GFP) heart; white asterisks indicate endocardial touchdowns; numbers in the bottom right corner refer to seconds. All images are ventral views, anterior to the top. V, ventricle; A, atrium.

The online version of this article includes the following video and figure supplement(s) for figure 1:

**Figure supplement 1.** Endocardial protrusions in the ventricle are mostly located in the outer curvature and are close to delaminating or trabecular cardiomyocytes (CMs).

**Figure 1—video 1.** Endocardial touchdowns during cardiac contraction.

https://elifesciences.org/articles/73231/figures#fig1video1

**Figure 1—video 2.** Endocardial protrusions extend along delaminating cardiomyocytes (CMs) at 60 hours post-fertilization (hpf).

https://elifesciences.org/articles/73231/figures#fig1video2

**Figure 1—video 3.** Endocardial protrusions are in close proximity to trabecular cardiomyocytes (CMs) at 72 hours post-fertilization (hpf).

https://elifesciences.org/articles/73231/figures#fig1video3

trabeculae, 'finger-like' multicellular projections, starting at 72 hpf (*Figure 1D, D' and D''''*). At this stage, we observed that endocardial protrusions can also be detected in close proximity to trabecular CMs (*Figure 1D'' and D'''*, *Figure 1—video 3*). Cardiac trabeculae are mostly present in the outer curvature of the ventricle at early developmental stages in zebrafish (*Liu et al., 2010*). Interestingly, we observed a spatial correlation between endocardial protrusions and trabeculation. At 48 hpf, endocardial protrusions are mostly located in the outer curvature of the ventricle (*Figure 1—figure supplement 1C, F*). At 60 and 72 hpf, respectively, 79% and 83% of all endocardial protrusions in the ventricle are located in the outer curvature (*Figure 1—figure supplement 1D-F*); 69% of all endo-cardial protrusions in the outer curvature are close to delaminating CMs at 60 hpf, and 91% of all endocardial protrusions in the outer curvature are close to trabecular CMs at 72 hpf (*Figure 1—figure supplement 1D, E, G*). Moreover, 98% of delaminating CMs and 93% of trabecular CMs are in close proximity to endocardial protrusions at 60 and 72 hpf, respectively (*Figure 1—figure supplement 1D, E, H*). Together these data lead us to speculate that endocardial protrusions play a role during endocardium-myocardium interactions.

## Genetically blocking endocardial protrusion formation reduces myocardial trabeculation

Since we observed a correlation between the presence of endocardial protrusions and myocardial trabeculation, we next aimed to examine the function of endocardial protrusions during cardiac morphogenesis. To this aim, we generated a transgenic line, *Tg(UAS: irsp53$^{dn}$-p2a-RFP)*, to specifically block protrusion formation in the endothelium. IRSp53 regulates the actin cytoskeleton to enable cells to form different types of membrane extensions (*Nakagawa et al., 2003*; *Millard et al., 2005*; *Scita et al., 2008*). By crossing the *Tg(UAS: irsp53$^{dn}$-p2a-RFP)* line to the *TgBAC(cdh5:Gal4ff)* line to over-express Irsp53$^{dn}$ specifically in ECs, we observed a 70% reduction in the number of endocardial protru-sions at 48 hpf (*Figure 2A, B and E*), while their distribution appeared mostly unaffected (*Figure 2A, B and F*). To test the hypothesis that endocardial protrusions modulate myocardial trabeculation, we analyzed embryos overexpressing *irsp53$^{dn}$* in their ECs in a CM membrane line (*Tg(myl7:BFP-CAAX)*). Upon *irsp53$^{dn}$* overexpression in ECs, we detected fewer endocardial touchdowns (*Figure 2A and B*), as well as a reduction in cardiac trabeculation (*Figure 2C, D, G and G'*). In order to identify a possible effect of endocardial protrusions on CM proliferation, we overexpressed *irsp53$^{dn}$* in the endothe-lium in the context of the *Tg(myl7:mVenus-gmnn)* reporter to visualize cycling CMs. Compared with controls, endothelial overexpression of *irsp53$^{dn}$* led to significantly fewer mVenus-Gmnn$^+$ CMs in the ventricle (*Figure 2H and I*).

## Apelin signaling positively regulates endocardial protrusion formation and myocardial trabeculation

We have recently found that Apelin signaling regulates endothelial filopodia formation during angio-genesis in the zebrafish trunk (*Helker et al., 2020*). Therefore, we hypothesized that Apelin signaling

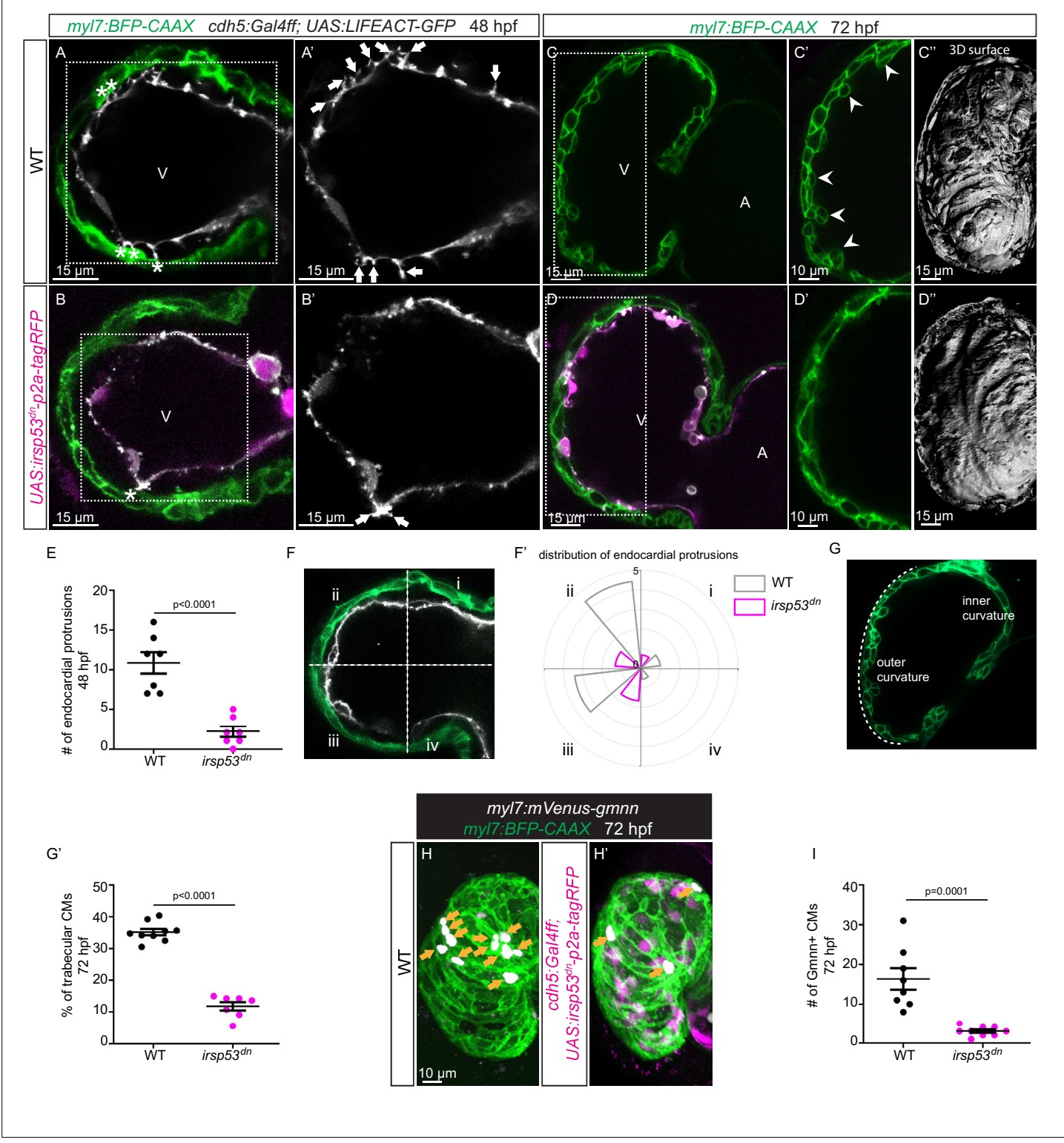

**Figure 2.** Blocking endocardial protrusion formation reduces cardiac trabeculation. (**A–D**) Confocal projection images of the heart of *Tg(myl7:BFP-CAAX); Tg(cdh5:Gal4ff); Tg(UAS:LIFEACT-GFP);±Tg(UAS:irsp53^dn^-p2a-tagRFP)* zebrafish at 48 (**A–B**) and 72 (**C–D**) hours post-fertilization (hpf). (**A–B**) Endocardial protrusions (white arrows) and touchdowns (white asterisks) are reduced in embryos with endothelial overexpression of *irsp53^dn^*. (**C–D**) Cardiac trabeculation (arrowheads) is reduced in larvae with endothelial overexpression of *irsp53^dn^*; (**C'–D**) 3D rendering. (**E**) Quantification of the number of endocardial protrusions in wild-type and in embryos with endothelial overexpression of *irsp53^dn^* at 48 hpf. (**F-F'**) Illustration of the division of the 48 hpf ventricle into four regions (**F**). Distribution and average number of endocardial protrusions in different regions of mid-sagittal sections of

*Figure 2 continued on next page*

*Figure 2 continued*

the ventricle from 48 hpf wild-type and *irsp53*[dn] embryos (**F'**). (**G–G'**) Illustration of the division of the 72 hpf ventricle into the outer and inner curvature (**G**). Quantification of the percentage of trabecular cardiomyocytes (CMs) in the outer curvature of wild-type and *irsp53*[dn] larvae at 72 hpf (**G'**). (**H–H'**) 72 hpf larvae with endothelial overexpression of *irsp53*[dn] display a reduced number of *myl7*:mVenus-Gmnn⁺ CMs (yellow arrows) in their ventricle. (**I**) Quantification of the number of mVenus-Gmnn⁺ CMs in the ventricle of wild-type and *irsp53*[dn] larvae at 72 hpf. All images are ventral views, anterior to the top. V, ventricle; A, atrium. Data in graphs expressed as mean ± SEM.

also regulates endocardial protrusion formation. In order to visualize the expression of *apln* and *aplnrb* at single-cell resolution in the heart, we examined the *TgBAC(apln:EGFP)* reporter line (*Helker et al., 2020*) and generated a novel *Tg(aplnrb:VenusPEST)* reporter line. We detected *apln*:EGFP expression in the myocardium at 48 and 72 hpf (*Figure 3A and B*). Furthermore, we detected *aplnrb*:VenusPEST expression in the endocardium at 48 and 72 hpf (*Figure 3C and D*). These results suggest that *apln* is expressed in the myocardium while *aplnrb* is expressed in EdCs. Based on these results, we hypothesized that Apelin signaling plays a role during endocardium-myocardium interactions.

To test this hypothesis, we used mutants for *aplnra* (*Helker et al., 2015*), *aplnrb* (*Helker et al., 2015*), *apln* (*Helker et al., 2015*), and *apela* (*Chng et al., 2013*). Since *apela* mutants fail to form a heart (*Chng et al., 2013*), we did not analyze them. While most *aplnrb* mutants fail to form a heart (*D'Amico et al., 2007*; *Zeng et al., 2007*), a low number of them do. By analyzing *aplnrb* mutants that do form a heart, we observed that they exhibit a reduced number of endocardial protrusions at 48 hpf (*Figure 4—figure supplement 1A, B*) and a reduced number of trabeculae at 72 hpf (*Figure 4—figure supplement 1C, D*). In wild-type embryos, the CJ between the endocardium and myocardium in the outer curvature of the ventricle appears to be mostly degraded at 72 hpf (*Figure 4—figure supplement 1C*); however, the CJ in *aplnrb* mutants appears to be thicker at this stage (*Figure 4—figure supplement 1D*). In addition, *aplnra* mutants also exhibit a reduced number of trabeculae at 72 hpf (*Figure 4—figure supplement 1E* and F). We further observed that *apln* mutants exhibit a significantly lower number of endocardial protrusions at 24 and 48 hpf (*Figure 4A–D–*). In line with fewer endocardial protrusions, *apln* mutants also exhibit a reduced number of endocardial touchdowns at 48 hpf (*Figure 4C and D*), and a reduced number of trabeculae at 72 hpf (*Figure 4E, F and J*). Altogether, these results indicate that Apelin signaling regulates endocardial protrusion formation and myocardial trabeculation.

To further examine the function of Apelin-dependent endocardial protrusions during cardiac trabeculation, we next analyzed CM proliferation in *apln* mutants using EdU labeling. Homozygous *apln* mutants exhibit a significantly decreased number of EdU⁺ CMs in their ventricle compared with *apln*⁺/⁺ siblings (*Figure 4—figure supplement 2*). In addition, *apln* mutants also display a significantly thicker CJ compared with *apln*⁺/⁺ siblings at 72 hpf (*Figure 4C–F, K and L*). However, we did not observe any obvious defects in sarcomere formation (*Figure 4—figure supplement 3A* and B), heart rate, ejection fraction, or blood circulation in *apln* mutants (*Figure 4—figure supplement 3C* and D; *Figure 4—videos 1–2*), indicating that the myocardial trabeculation phenotype is caused by the endocardial protrusion defect and is not secondary to cardiac dysfunction.

Notch signaling negatively regulates endothelial sprouting and protrusion formation in several vascular beds (*Hellström et al., 2007*; *Leslie et al., 2007*; *Siekmann and Lawson, 2007*; *Suchting et al., 2007*). In order to determine whether Notch signaling also regulates endocardial protrusion formation, we treated embryos with the γ-secretase inhibitor RO4929097 and observed a decrease in Notch reporter expression in the endocardium (*Figure 4—figure supplement 4A* and B) as well as an increased number of endocardial protrusions in the ventricle (*Figure 4—figure supplement 4E, F, and H*). However, and in line with the touchdown reduction phenotype in Notch-deficient mice, we observed a reduction in endocardial touchdowns in Notch inhibitor treated zebrafish larvae at 48 hpf (*Figure 4—figure supplement 4C, D, and G*).

Altogether, these results indicate that myocardial derived Apelin promotes endocardial protrusion formation while Notch signaling inhibits it. Furthermore, Apelin signaling is required for cardiac trabeculation, possibly via the formation of endocardial protrusions.

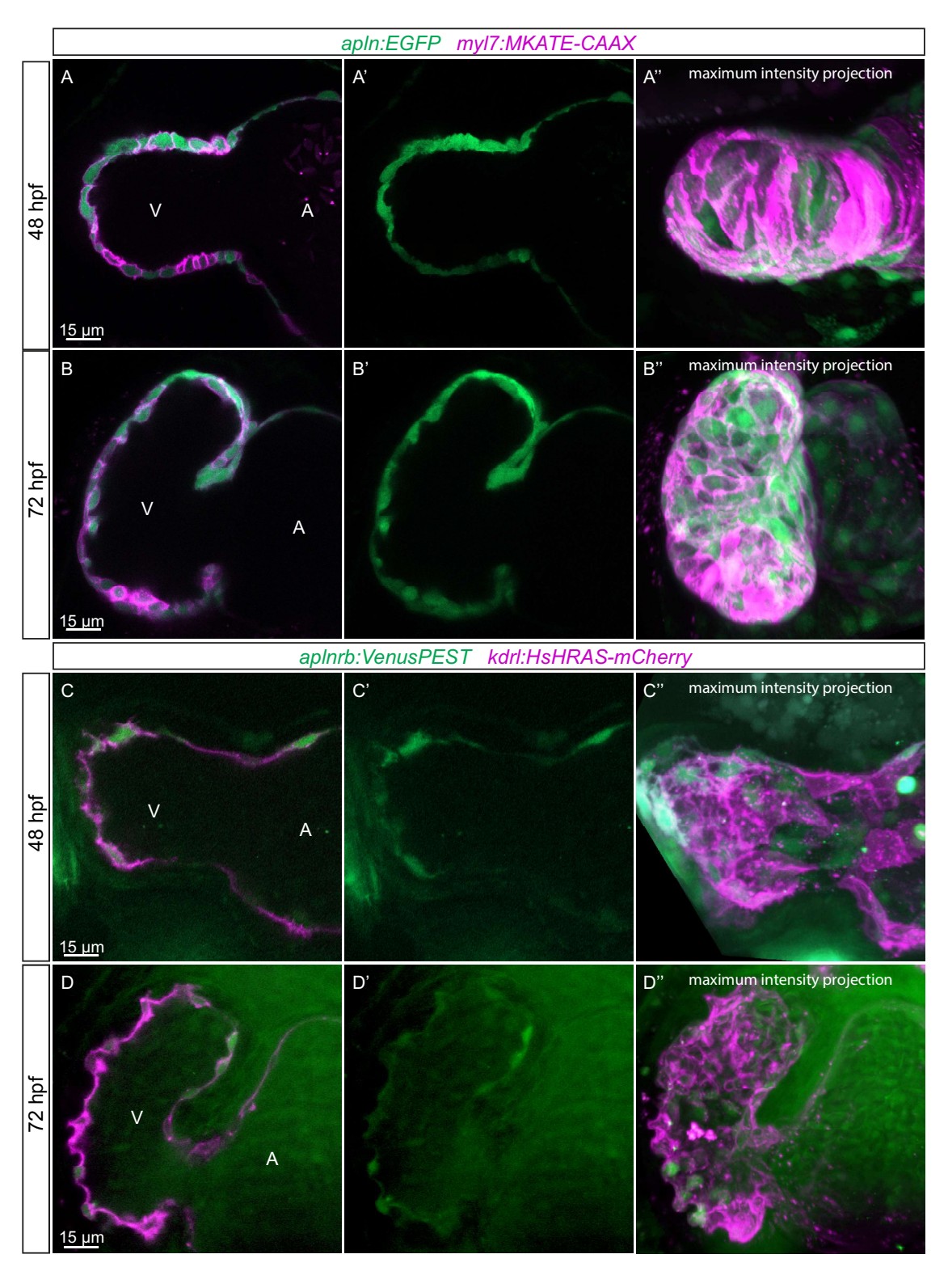

**Figure 3.** Expression pattern of Apelin signaling pathway components. (**A–D**) Confocal projection images of the heart of *TgBAC(apln:EGFP); Tg(myl7:MKATE-CAAX)* (**A, B**) and *TgBAC(aplnrb:VenusPEST); Tg(kdrl:HsHRAS-mCherry)* (**C, D**) zebrafish at 48 (**A, C**) and 72 (**B, D**) hours post-fertilization (hpf). (**A′–′D′**) Maximum intensity projections. (**A–B**) *TgBAC(apln:*EGFP) expression is detectable in the myocardium at 48 (**A**) and 72 (**B**) hpf. (**C–D**) *TgBAC(aplnrb:*VenusPEST) expression is detectable in the endocardium with higher expression in the ventricular endocardium at 48 (**C**) and 72 (**D**) hpf. All images are ventral views, anterior to the top. V, ventricle; A, atrium.

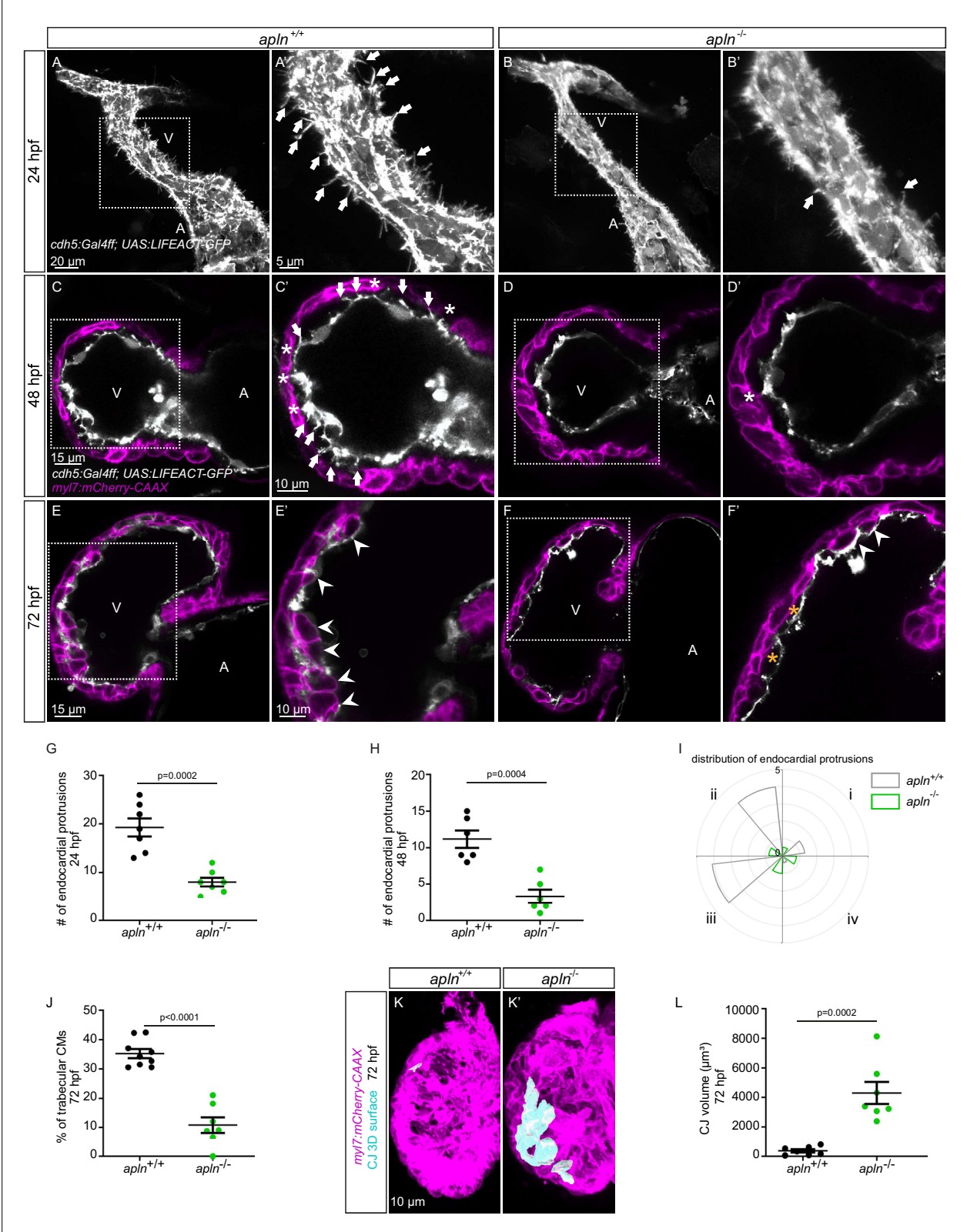

**Figure 4.** Loss of Apelin signaling leads to reduced endocardial protrusion and reduced myocardial trabeculation. (**A–F**) Confocal projection images of the heart of *Tg(cdh5:Gal4ff); Tg(UAS:LIFEACT-GFP)* zebrafish at 24 hours post-fertilization (hpf) (**A–B**) and of the heart of *Tg(myl7:mCherry-CAAX); Tg (cdh5:Gal4ff); Tg(UAS:LIFEACT-GFP)* (**C–F**) zebrafish at 48 (**C–D**) and 72 (**E–F**) hpf. Maximum intensity projections (**A–B**) and mid-sagittal sections (**C–F**). (**A**) Endocardial protrusions (arrows) in *apln+/+* embryos at 24 hpf. (**B**) The number of endocardial protrusions (arrows) is reduced in *apln-/-* siblings at 24

*Figure 4 continued on next page*

*Figure 4 continued*

hpf. (**C–D**) The numbers of endocardial protrusions (arrows) and touchdowns (white asterisks) are reduced in *apln-/-* embryos (**D**) at 48 hpf compared with *apln+/+* siblings (**C**). (**E–F**) *apln-/-* larvae (**F**) exhibit reduced trabeculation (arrowheads) and thicker cardiac jelly (CJ) (yellow asterisks) at 72 hpf compared with *apln+/+* siblings (**E**). (**G–H**) Quantification of the number of endocardial protrusions in the ventricle of *apln+/+* and *apln-/-* siblings at 24 (**G**) and 48 (**H**) hpf. (**I**) Distribution and average number of endocardial protrusions in different regions of mid-sagittal sections of the ventricle from 48 hpf *apln+/+* and *apln-/-* siblings. (**J**) Quantification of the percentage of trabecular cardiomyocytes (CMs) in the outer curvature of *apln+/+* and *apln-/-* siblings at 72 hpf. (**K–K'**) Maximum intensity projections. *apln-/-* larvae (**K'**) exhibit a thicker CJ at 72 hpf compared with *apln+/+* siblings (**K**). (**L**) Quantification of the CJ volume in the outer curvature of *apln+/+* and *apln-/-* siblings at 72 hpf. All images are ventral views, anterior to the top. V, ventricle; A, atrium; +/+, *apln+/+*; -/-, *apln-/-*. Data in graphs expressed as mean ± SEM.

The online version of this article includes the following video and figure supplement(s) for figure 4:

**Figure supplement 1.** *aplnrb* mutants exhibit reduced endocardial protrusion formation and reduced trabeculation, and *aplnra* mutant exhibit a mild reduction in trabeculation.

**Figure supplement 2.** Apelin signaling regulates cardiomyocyte (CM) proliferation in the ventricle.

**Figure supplement 3.** Wild-type like sarcomere structure and heart function in *apln-/-* larvae.

**Figure supplement 4.** Notch signaling represses endocardial protrusion formation.

**Figure 4—video 1.** *apln+/+* blood circulation at 48 hours post-fertilization (hpf).
https://elifesciences.org/articles/73231/figures#fig4video1

**Figure 4—video 2.** *apln-/-* blood circulation at 48 hours post-fertilization (hpf).
https://elifesciences.org/articles/73231/figures#fig4video2

## The function of endocardial *Nrg2a* in trabeculation requires endocardial protrusions

Nrg-ErbB signaling is indispensable for cardiac trabeculation in mouse and zebrafish (*Gassmann et al., 1995*; *Lee et al., 1995*; *Meyer and Birchmeier, 1995*; *Lai et al., 2010*; *Liu et al., 2010*; *Peshkovsky et al., 2011*; *Rasouli and Stainier, 2017*). To determine whether endocardial protrusions are required for Nrg-ErbB signaling, we first overexpressed *nrg2a* in the endothelium using the *Tg(fli1a:nrg2a-p2a-tdTomato)* line (*Rasouli and Stainier, 2017*). Overexpression of *nrg2a* in the endothelium results in hypertrabeculation as well as a multilayered myocardium (*Figure 5A, B and E–G*; *Figure 5—figure supplement 1A and B*). Strikingly, overexpressing *nrg2a* in the endothelium while blocking endocardial protrusion formation by endothelial overexpression of *irsp53dn* is not sufficient to restore cardiac trabeculation or induce CM multilayering (*Figure 5C-C' and E-G*). In line with these results, overexpressing *nrg2a* in the endothelium of homozygous *apln* mutants is not sufficient to restore cardiac trabeculation or induce CM multilayering (*Figure 5D-D' and E-G*). Importantly, we did not detect a change in the expression levels of *nrg2a* in *apln* mutant hearts at 48 hpf (*Figure 5—figure supplement 1C*). Taken together, these data indicate that endocardial protrusions are required for Nrg-ErbB signaling during cardiac trabeculation.

## Genetically blocking endocardial protrusion formation attenuates Erk signaling in cardiomyocytes

An important molecule in the Nrg/ErbB signaling pathway is the extracellular signal-regulated kinase Erk (*Lai et al., 2010*). In order to visualize Erk activity in CMs in living zebrafish, as a readout of ErbB signaling, we generated novel reporter lines (*Tg(myl7:ERK-KTR-Clover-p2a-H2B-tagBFP)* and *Tg(myl7:ERK-KTR-Clover-p2a-H2B-mScarlet)*) that use the kinase translocation reporter (KTR) technology (*Regot et al., 2014*; *de la Cova et al., 2017*). When Erk is inactive, the KTR is unphosphorylated and Clover can be detected in the nucleus; in contrast, when Erk is active, the KTR is phosphorylated and Clover can be detected in the cytoplasm (*de la Cova et al., 2017*). We observed that most ventricular CMs in wild-type larvae display active Erk signaling with cytoplasmic Clover expression (*Figure 6A*). Treating embryos expressing the reporter with a MEK inhibitor led to an increased number of ventricular CMs with nuclear Clover expression (i.e., inactive Erk signaling) indicating that the reporter is functional (*Figure 6—figure supplement 1*). Next, we treated embryos expressing the reporter with an ErbB2 inhibitor and found an increased number of ventricular CMs with nuclear Clover expression (*Figure 6B*). To determine whether endocardial protrusions modulate myocardial Erk signaling activity, we genetically blocked endocardial protrusion formation via endothelial overexpression of *irsp53dn* and observed more ventricular CMs with nuclear Clover expression

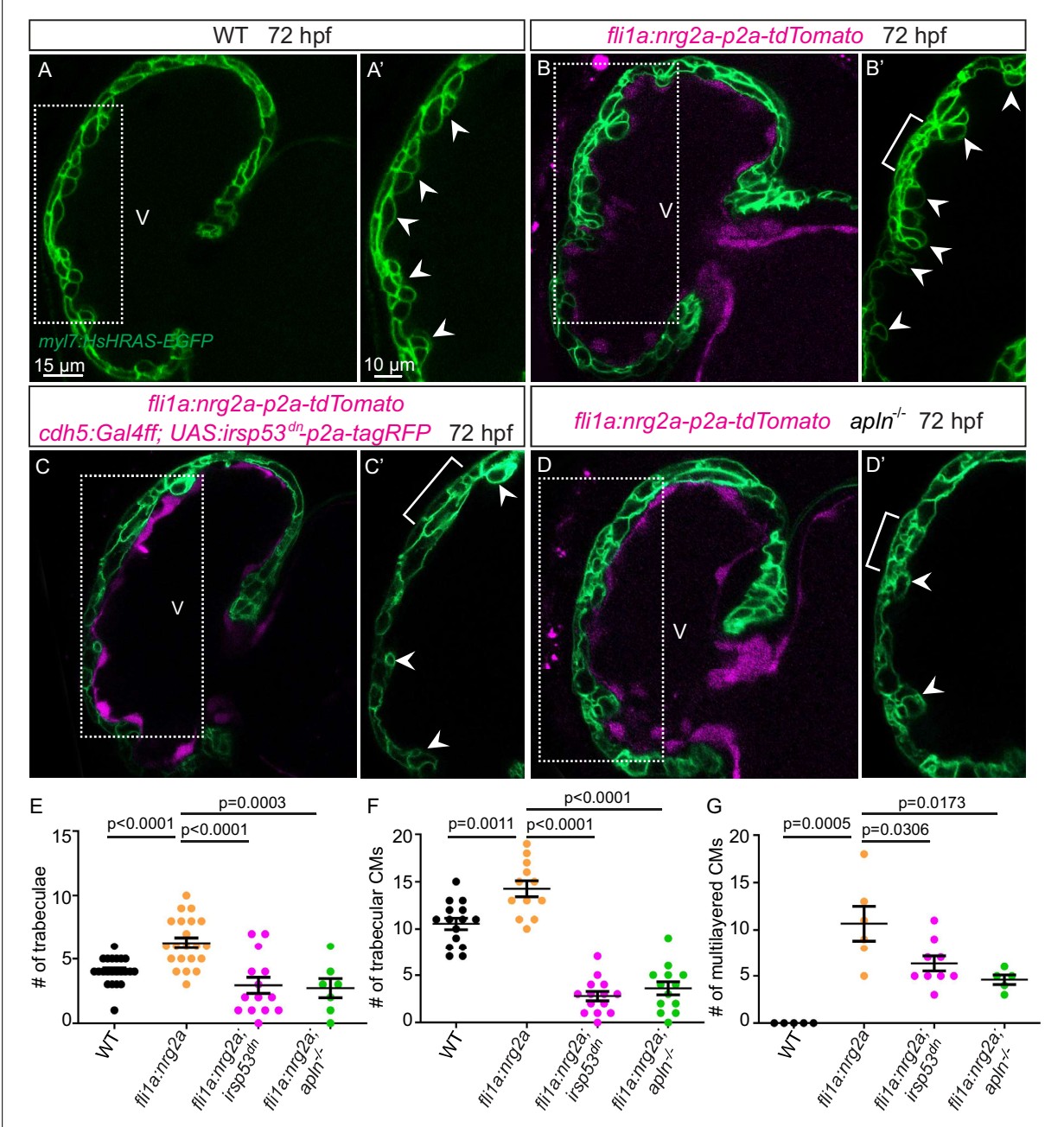

**Figure 5.** Endocardial protrusions are necessary for *nrg2a* function. (**A–D**) Confocal projection images of the heart of *Tg(myl7:HsHRAS-EGFP)* larvae at 72 hours post-fertilization (hpf). (**A–B**) Overexpression of *nrg2a* in the endothelium (**B**) leads to an increased number of trabeculae (arrowheads) and the multilayering of cardiomyocytes (CMs) (brackets) compared with wild-type (**A**). (**C**) Larvae with endothelial overexpression of *nrg2a* and *irsp53*$^{dn}$ exhibit a reduced number of trabeculae (arrowheads) and of multilayered CMs (brackets) compared with larvae with endothelial overexpression of *nrg2a* alone (**B**). (**D**) *apln* mutant larvae with endothelial overexpression of *nrg2a* exhibit a reduced number of trabeculae (arrowheads) and of multilayered CMs (brackets) compared with wild-type larvae with endothelial overexpression of *nrg2a* (**B**). (**E**) Quantification of the number of trabeculae. (**F**) Quantification of the number of trabecular CMs. (**G**) Quantification of the number of multilayered CMs in the ventricle. Brackets indicate multilayered CMs. All images are ventral views, anterior to the top. V, ventricle. Data in graphs expressed as mean ± SEM.

The online version of this article includes the following figure supplement(s) for figure 5:

**Figure supplement 1.** *nrg2a* expression does not appear to be affected in *apln* mutants.

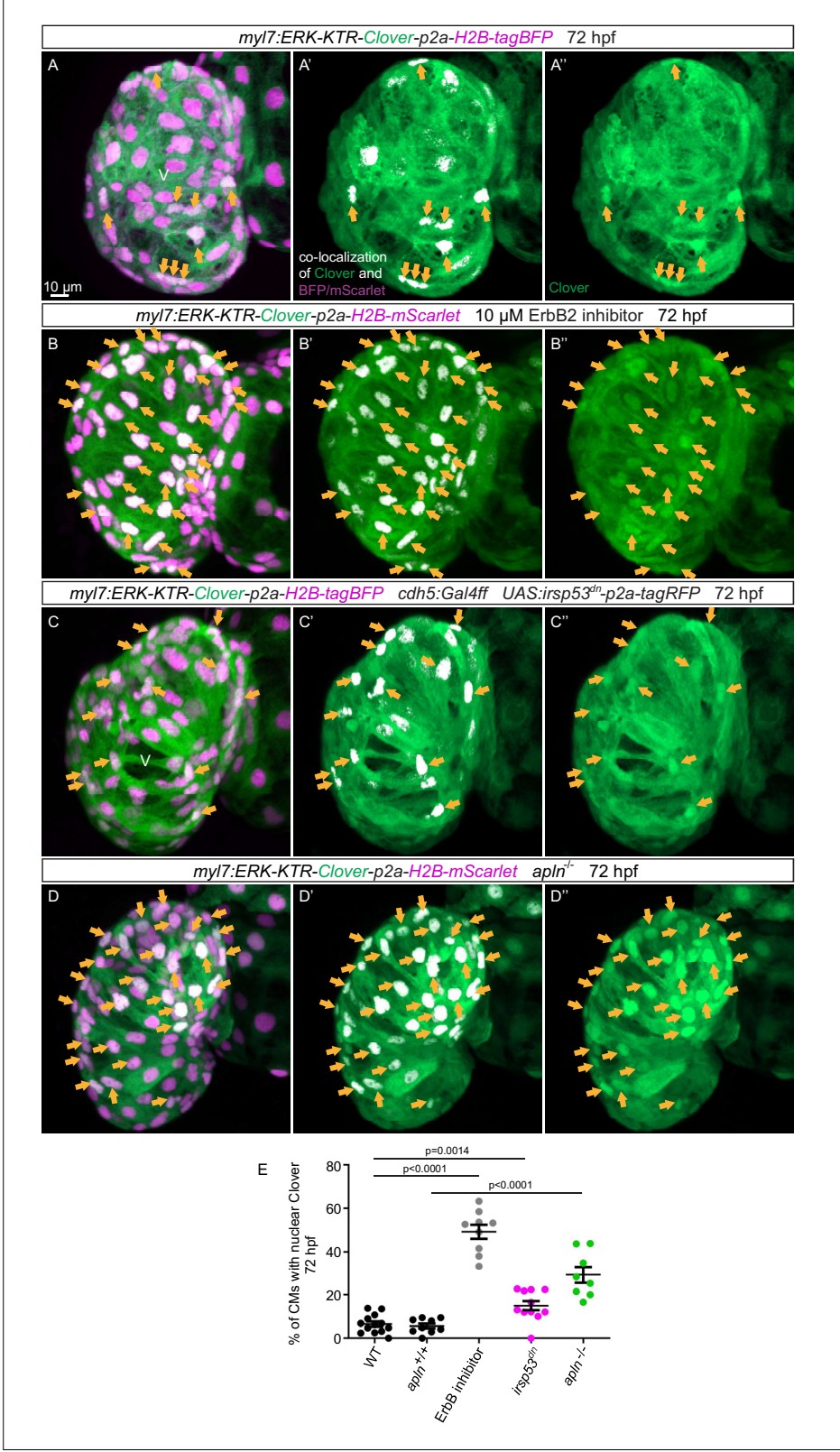

**Figure 6.** Blocking endocardial protrusion formation reduces myocardial extracellular signal-regulated kinase (Erk) signaling activity. (**A–D**) Maximum intensity projections of confocal images of the heart of *Tg(myl7:ERK-KTR-Clover-p2a-H2B-tagBFP/mScarlet)* larvae at 72 hours post-fertilization (hpf). (**A**) Visualization of Erk activity by a cardiomyocyte (CM)-specific ERK-kinase translocation reporter (KTR) reporter. Nuclear Clover expression

*Figure 6 continued on next page*

*Figure 6 continued*

(arrows) indicates CMs with inactive Erk signaling. (**B**) Larvae treated with an ErbB2 inhibitor exhibit an increased number of CMs with inactive Erk signaling (arrows) compared with control larvae (**A**). (**C**) Larvae with endothelial overexpression of *irsp53$^{dn}$* exhibit an increased number of CMs with inactive Erk signaling (arrows) compared with control larvae (**A**). (**D**) *apln* mutant larvae exhibit an increased number of CMs with inactive Erk signaling (arrows) compared with *apln$^{+/+}$* siblings. (**E**) Quantification of the percentage of ventricular CMs with nuclear Clover expression. All images are ventral views, anterior to the top. V, ventricle. Data in graphs expressed as mean ± SEM.

The online version of this article includes the following figure supplement(s) for figure 6:

**Figure supplement 1.** Validation of the extracellular signal-regulated kinase (Erk) reporter line using MEK inhibitor treatment.

**Figure supplement 2.** Schematic model.

---

(*Figure 6C and E*) compared with control (*Figure 6A and E*), indicating more ventricular CMs with inactive Erk signaling. In line with these results, we observed more ventricular CMs with inactive Erk signaling in homozygous *apln* mutants (*Figure 6D and E*) compared with *apln$^{+/+}$* siblings (*Figure 6A and E*). Altogether, these observations indicate that Apelin signaling-dependent endocardial protrusions promote Nrg/ErbB/Erk signaling in CMs.

## Discussion
### Endocardial protrusions are required for trabeculation

Cardiac trabeculation is initiated, at least in zebrafish, by individual CMs delaminating from the myocardial monolayer and protruding into the lumen (*Liu et al., 2010*; *Peshkovsky et al., 2011*; *Staudt et al., 2014*; *Jiménez-Amilburu et al., 2016*; *Priya et al., 2020*). In contrast, the myocardium in mouse is multilayered at the onset of trabeculation. Several studies have reported that the endocardium plays an important role during cardiac trabeculation (*Grego-Bessa et al., 2007*; *Lai et al., 2010*; *D'Amato et al., 2016*; *Rasouli and Stainier, 2017*; *Del Monte-Nieto et al., 2018*; *Qu et al., 2019*). Furthermore, it has recently been shown that EdCs, similar to ECs, undergo sprouting (*Del Monte-Nieto et al., 2018*). However, in comparison with endothelial sprouting, little is known about the morphogenetic events underlying endocardial sprouting and their effect on cardiac morphogenesis including trabeculation.

In mouse, endocardial sprouting and touchdown formation occur early during cardiac trabeculation (*Del Monte-Nieto et al., 2018*). These observations are in line with our data in zebrafish, suggesting that the morphogenetic events of cardiac trabeculation are evolutionarily conserved. However, endocardial sprouts in mouse appear to be cellular (*Del Monte-Nieto et al., 2018*), whereas endocardial protrusions in zebrafish appear more similar to filopodia, although these differences may be due to the fact that fixed tissue was used for the mouse work compared with live tissue for the zebrafish work. CM delamination and trabeculation occur in the outer curvature of the ventricle (*Liu et al., 2010*; *Jiménez-Amilburu et al., 2016*; *Rasouli and Stainier, 2017*). Consistent with these observations, we find that endocardial protrusions are mostly located in the outer curvature of the ventricle and extend along delaminating CMs as well as trabecular CMs. The temporal and spatial correlation between the emergence and location of endocardial protrusions and CM delamination therefore suggests a role for endocardial protrusions in cardiac trabeculation.

### Molecular regulators of endocardial sprouting

During sprouting angiogenesis, so-called tip cells lead the new sprouts (*Gerhardt, 2008*). Tip cells dynamically extend filopodia to identify growth factors in their environment (*Gerhardt, 2008*). Apelin and Notch signaling have been previously identified as regulators of endothelial filopodia formation (*Hellström et al., 2007*; *Suchting et al., 2007*; *Helker et al., 2020*). In contrast, the pathways regulating endocardial sprouting are largely unknown. Only Tie2 signaling has been identified to date as a regulator of endocardial sprouting, and *Tie2*-deficient mice exhibit fewer endocardial touchdowns (*Qu et al., 2019*). We have recently shown that Apelin signaling regulates filopodia formation during sprouting angiogenesis in the trunk (*Helker et al., 2020*). In line with these published observations, we now show that Apelin regulates endocardial filopodia formation and endocardial sprouting

(*Figure 6—figure supplement 2*), revealing a conserved role for Apelin signaling during endothelial and endocardial sprouting.

Consistent with the negative regulation of sprouting angiogenesis by Notch signaling in ECs (*Hellström et al., 2007*; *Leslie et al., 2007*; *Siekmann and Lawson, 2007*; *Suchting et al., 2007*), we found that Notch signaling also negatively regulates endocardial protrusion formation. Interestingly, inhibition of Notch signaling also leads to an increased number of delaminated CMs and trabeculae, although Notch signaling may also affect CM behavior cell autonomously (*Han et al., 2016*; *Priya et al., 2020*).

## Endocardium-myocardium communication is essential for trabeculation

Cell communication between the endocardial and myocardial cells is required for cardiac trabeculation (*Gassmann et al., 1995*; *Lee et al., 1995*; *Meyer and Birchmeier, 1995*; *Lai et al., 2010*; *Liu et al., 2010*; *Rasouli and Stainier, 2017*; *Gunawan et al., 2021*). Several studies have shown that endocardial-derived Nrg is required to activate ErbB receptor complexes on CMs (*Gassmann et al., 1995*; *Meyer and Birchmeier, 1995*; *Grego-Bessa et al., 2007*; *Rasouli and Stainier, 2017*). In zebrafish, *nrg2a*, which is expressed in the developing endocardium, is required for trabeculation (*Rasouli and Stainier, 2017*), and *nrg2a* overexpression in the endothelium results in an increased number of trabeculae and trabecular CMs (*Figure 5*), indicating that endocardial *nrg2a* is necessary and sufficient for trabeculation. Notably, blocking endocardial protrusion formation compromises the function of endocardial *nrg2a* during trabeculation, suggesting that endocardial protrusions are required for *nrg2a* function during trabeculation. The membrane-bound form of Nrg2a has a molecular weight of 55 kD, and after cleavage, the presumed secreted form has a molecular weight of 22 kD. In contrast, the most active form of Apelin is 13 amino acids long, with a molecular weight of 1.52 kD. Therefore, it is possible that due to its small size, Apelin is able to diffuse through the CJ, while Nrg2a needs to be transported by endocardial protrusions toward the myocardium.

Like other receptor tyrosine kinases, ErbB receptors activate multiple signaling cascades, including the MAPK cascade, upon ligand stimulation, leading to the phosphorylation of ERK1/2 (*Sweeney et al., 2001*; *Wee and Wang, 2017*). Accordingly, attenuated ERK phosphorylation is observed in CMs in Nrg1/ErbB signaling-deficient mice (*Lai et al., 2010*). By analyzing a novel reporter of Erk activity in CMs, we observed that the inhibition of endocardial protrusion formation, as well as the genetic inactivation of Apelin signaling, leads to attenuated Erk phosphorylation in CMs. Altogether, these data suggest that Apelin signaling-dependent endocardial protrusions modulate ErbB signaling in CMs (*Figure 6—figure supplement 2*).

It has recently been shown that EC filopodia modulate neurogenesis by affecting progenitor cell proliferation in the developing brain of mice and zebrafish (*Di Marco et al., 2020*; *Taberner et al., 2020*). Of interest, ErbB signaling is also known for its function within the nervous system (*Buonanno and Fischbach, 2001*). Thus, one might speculate that Nrg/ErbB signaling also plays a role during the modulation of neurogenesis by endothelial filopodia. Several studies have reported cell-to-cell communication by cytonemes in different animal models (*Ramírez-Weber and Kornberg, 1999*; *Holzer et al., 2012*; *Luz et al., 2014*; *Pröls et al., 2015*). Whether endocardial protrusions qualify as cytonemes needs further analysis. However, our data indicate that Apelin-dependent endocardial protrusions are required for the communication between endocardial and myocardial cells via Nrg/ErbB signaling (*Figure 6—figure supplement 2*).

In summary, our work describes how endocardial sprouting is required for Nrg/ErbB signaling during cardiac trabeculation. Furthermore, we identify Apelin signaling as a positive regulator of endocardial sprouting.

## Materials and methods

**Key resources table**

| Reagent type (species) or resource | Designation | Source or reference | Identifiers | Additional information |
|---|---|---|---|---|
| Genetic reagent (*Danio rerio*) | *TgBAC(apln:EGFP)^bns157* | *Helker et al., 2020* | ZFIN: *bns157* | |
| Genetic reagent (*Danio rerio*) | *TgBAC(cdh5:Gal4ff)^mu101* | *Bussmann et al., 2011* | ZFIN: *mu101* | |

*Continued on next page*

*Continued*

| Reagent type (species) or resource | Designation | Source or reference | Identifiers | Additional information |
|---|---|---|---|---|
| Genetic reagent (*Danio rerio*) | *Tg(UAS:LIFEACT-GFP)$^{mu271}$* | *Helker et al., 2013* | ZFIN: *mu271* | |
| Genetic reagent (*Danio rerio*) | *Tg(fli1a:nrg2a-p2a-tdTomato)$^{bns199}$* | *Rasouli and Stainier, 2017* | ZFIN: *bns199* | |
| Genetic reagent (*Danio rerio*) | *Tg(myl7:mCherry-CAAX)$^{bns7}$* | *Uribe et al., 2018* | ZFIN: *bns7* | |
| Genetic reagent (*Danio rerio*) | *Tg(myl7:BFP-CAAX)$^{bns193}$* | *Guerra et al., 2018* | ZFIN: *bns193* | |
| Genetic reagent (*Danio rerio*) | *Tg(myl7:MKATE-CAAX)$^{sd11}$* | *Lin et al., 2012* | ZFIN: *sd11* | |
| Genetic reagent (*Danio rerio*) | *Tg(kdrl:HsHRAS-mCherry)$^{s896}$* | *Chi et al., 2008* | ZFIN: *s896* | |
| Genetic reagent (*Danio rerio*) | *Tg(myl7:HRAS-EGFP)$^{s883}$* | *D'Amico et al., 2007* | ZFIN: *s883* | |
| Genetic reagent (*Danio rerio*) | *Tg(tp1-MmHbb:EGFP)$^{um14}$* | *Parsons et al., 2009* | ZFIN: *um14* | |
| Genetic reagent (*Danio rerio*) | *Tg(myl7:mVenus-gmnn)$^{ncv43Tg}$* | *Jiménez-Amilburu et al., 2016* | ZFIN: *ncv43Tg* | |
| Genetic reagent (*Danio rerio*) | *Tg(UAS: irsp53$^{dn}$-p2a-tagRFP)$^{bns440}$* | This paper | *bns440* | See Materials and methods section |
| Genetic reagent (*Danio rerio*) | *TgBAC(aplnrb:VenusPEST)$^{mr13}$* | This paper | *mr13* | See Materials and methods section |
| Genetic reagent (*Danio rerio*) | *Tg(–0.8myl7:ERK-KTR-Clover-p2a-Hsa.H2B-tagBFP)* | This paper | | See Materials and methods section |
| Genetic reagent (*Danio rerio*) | *Tg(–0.8myl7:ERK-KTR-Clover-p2a-Hsa.H2B-mScarlet)$^{bns565}$* | This paper | *bns565* | See Materials and methods section |
| Genetic reagent (*Danio rerio*) | *apln$^{mu267}$* mutant | *Helker et al., 2015* | ZFIN: *mu267* | |
| Genetic reagent (*Danio rerio*) | *aplnrb$^{mu281}$* mutant | *Helker et al., 2015* | ZFIN: *mu281* | |
| Genetic reagent (*Danio rerio*) | *aplnra$^{mu296}$* mutant | *Helker et al., 2015* | ZFIN: *mu296* | |
| Antibody | Alexa Fluor 488 anti-Chicken IgG (H + L) (goat polyclonal) | Thermo Fisher Scientific | Cat# A-11039 | (1:500) |
| Antibody | Alexa Fluor 568 anti-Mouse IgG (H + L) (goat polyclonal) | Thermo Fisher Scientific | Cat# A-11004 | (1:500) |
| Antibody | Alexa Fluor 647 anti-Rabbit IgG (H + L) (goat polyclonal) | Thermo Fisher Scientific | Cat# A-21244 | (1:500) |
| Antibody | Anti-GFP (chicken polyclonal) | AvesLab | Cat#: GFP-1020 | (1:500) |
| Antibody | Anti-mCherry (mouse monoclonal) | Takara Bio Clontech | Cat# 632,543 | (1:500) |
| Chemical compound, drug | Agarose, low gelling temperature | Sigma | A9414-25g | |
| Chemical compound, drug | EdU | Thermo Fisher Scientific | Cat# A10044 | (1 mM) |
| Chemical compound, drug | ErbB2 inhibitor PD168393 | Sigma | Cat# PZ0285 | (10 µM) |
| Chemical compound, drug | MEK inhibitor PD0325901 | Sigma | Cat# PZ0162 | (1 µM) |
| Chemical compound, drug | RO 4929097 | MedChemExpress | Cat# HY-11102 | (1 µM) |
| Other | DAPI | Sigma | Cat# D9542 | (1 µg/ml) |
| Commercial assay or kit | Alexa Fluor 568 Phalloidin | Thermo Fisher Scientific | Cat# A12380 | (1:100) |
| Commercial assay or kit | Click-iT EdU Cell Proliferation Kit for Imaging, Alexa Fluor 647 dye | Thermo Fisher Scientific | Cat# C10340 | |
| Commercial assay or kit | DyNAmo ColorFlash SYBR Green qPCR Mix | Thermo Fisher Scientific | Cat# F416S | |
| Commercial assay or kit | In-Fusion HD Cloning Plus | Takara Bio | Cat# 638,910 | |
| Commercial assay or kit | Maxima First Strand cDNA kit | Thermo Fisher Scientific | Cat# K1641 | |
| Commercial assay or kit | RNA clean and concentrator-5 | Zymo Research | R1016 | |
| Software, algorithm | Fiji Image J | *Schindelin et al., 2012* | RRID:SCR_002285 | |
| Software, algorithm | GraphPad Prism 8 | GraphPad Software | RRID:SCR_002798 | |
| Software, algorithm | Imaris – version 9.6.0 | Bitplane | RRID:SCR_007370 | |
| Software, algorithm | ZEN Digital Imaging | Zeiss | RRID:SCR_013672 | |

## Zebrafish lines

All zebrafish housing and husbandry were performed under standard conditions in accordance with institutional (Max Planck Society) and national ethical and animal welfare guidelines approved by the ethics committee for animal experiments at the Regierungspräsidium Darmstadt, Germany, as well as the FELASA guidelines (*Aleström et al., 2020*). Embryos were staged by hpf at 28.5°C (*Kimmel et al., 1995*).

Transgenic lines used in the study: *TgBAC(apln:EGFP)^bns157* (*Helker et al., 2020*), *TgBAC(cdh5:Gal4ff)^mu101* (*Bussmann et al., 2011*), *Tg(UAS:LIFEACT-GFP)^mu271* (*Helker et al., 2013*), *Tg(fli1a:nrg2a-p2a-tdTomato)^bns199* (*Rasouli and Stainier, 2017*), *Tg(myl7:mCherry-CAAX)^bns7* (*Uribe et al., 2018*), *Tg(myl7:BFP-CAAX)^bns193* (*Guerra et al., 2018*), *Tg(myl7:MKATE-CAAX)^sd11* (*Lin et al., 2012*), *Tg(kdrl:HsHRAS-mCherry)^s896* (*Chi et al., 2008*), *Tg(myl7:HRAS-EGFP)^s883* (*D'Amico et al., 2007*), *Tg(tp1-MmHbb:EGFP)^um14* (*Parsons et al., 2009*), *Tg(myl7:mVenus-gmnn)^ncv43Tg* (*Jiménez-Amilburu et al., 2016*), *Tg(UAS:irsp53^dn-p2a-tagRFP)^bns440* (this study), *TgBAC(aplnrb:VenusPEST)^mr13* (this study), *Tg(–0.8myl7:ERK-KTR-Clover-p2a-Hsa.H2B-tagBFP)* (this study, abbreviated as *Tg(myl7:ERK-KTR-Clover-p2a-H2B-tagBFP)*) and *Tg(–0.8myl7:ERK-KTR-Clover-p2a-Hsa.H2B-mScarlet)^bns565* (this study, abbreviated as *Tg(myl7:ERK-KTR-Clover-p2a-H2B-mScarlet)*).

Mutant lines used in the study: *apln^mu267* (*Helker et al., 2015*), *aplnrb^mu281* (*Helker et al., 2015*), *aplnra^mu296* (*Helker et al., 2015*).

## Generation of transgenic lines

To generate the *Tg(UAS:irsp53^dn-p2a-tagRFP)* line, a dominant negative form of *irsp53* (*Millard et al., 2005*; *Meyen et al., 2015*) was amplified by PCR using the primers: forward, 5'- TTCGAATTAGAT CTGTCGACCGCCACCATGTCTCGCACCGACGAGGT-3'; reverse, 5'- GTAGCTCCGCTTCCACGCGT CTGTGCAAAGCCTGCCATGC-3'. The amplification was cloned into a 5xUAS-p2a vector upstream of tagRFP.

To generate the *aplnrb* bacterial artificial chromosome (BAC) construct, we used the BAC clone CH211-102K containing the *aplnrb* locus. All recombineering steps were performed as described in *Bussmann and Schulte-Merker, 2011*, with the modifications as described in *Helker et al., 2019*. The following homology arms were used to generate the targeting PCR product of the Venus-pest_Kan cassette: aplnrb_HA1_GFP_fw: GAGCACATGACAAACAACTTCTCTGTGATCACTTCAAAGATTTT CTTGAAACCATGGTGAGCAAGGGCGAGGAG and aplnrb_HA2_kanR_rev: TCGTCGAAGTAATCTG GGCTATAGTCAGCAGTCATGTTGTCCATGGCATTTTCCAGAAGTAGTGAGGAG. The Kanamycin cassette was removed with a flippase.

To generate the *Tg(–0.8myl7:ERK-KTR-Clover-p2a-Hsa.H2B-tagBFP)* and *Tg(–0.8myl7:ERK-KTR-Clover-p2a-Hsa.H2B-mScarlet)* lines, ERK-KTR-Clover was amplified from addgene plasmid #59150 using forward primer: GCAAAGCAGACAGTGAACAAGCTTGCTAGCCCACCATGAAGGGCCGAA AGCCTCGGG and reverse primer: GTTAGTAGCTCCGCTTCCGTCGACGGCGGCGGTCACGAAC TCCAGCAGG. The PCR product was cloned into a Tol2 enabled vector containing p2a and H2B. Of note, no F1 adults were recovered for the *Tg(–0.8myl7:ERK-KTR-Clover-p2a-Hsa.H2B-tagBFP)* line.

All constructs were injected into AB embryos at the one-cell stage (30 pg/embryo) together with Tol2 mRNA (25 pg/embryo) to establish the line.

## Live imaging of stopped hearts

Zebrafish embryos and larvae were mounted in 1.4% low-melt agarose containing 1.6 mg/ml tricaine, to stop the heartbeat, on glass-bottom dishes. The samples were imaged with a Zeiss LSM800 confocal microscope using a 40×/1.1 W objective or a Zeiss LSM880 confocal microscope using a 20×/1.1 W objective.

## Live imaging of beating hearts

Zebrafish embryos were mounted in 0.8% low-melt agarose containing 0.2 mg/ml tricaine on glass-bottom dishes and kept in water containing 0.2 mg/ml tricaine through the experiment. Videos were acquired at 100 frames per second with a Hamamatsu ORCA flash 4.0 sCMOS camera and the image was binned 4 × 4 to achieve a calculated pixel resolution of 0.7 μm (40× objective).

## Inhibitor treatments

Zebrafish embryos were treated with 1 µM Notch inhibitor RO 4929097 for 24 hr (from 24 to 48 hpf), washed twice with egg water containing 0.1% (w/v) 1-phenyl-2-thiourea (PTU) and mounted in 1.4% low-melt agarose with 1.6 mg/ml tricaine for imaging.

Zebrafish embryos were treated with 1 µM MEK inhibitor PD 0325901 or 10 µM ErbB2 inhibitor PD 168393 from 56 to 72 hpf, washed twice with egg water containing 0.1% (w/v) PTU and mounted in 1.4% low-melt agarose with 1.6 mg/ml tricaine for imaging.

## EdU staining

Zebrafish embryos were treated with 1 mM EdU dissolved in 1% DMSO for 24 hr (from 48 to 72 hpf) or 44 hr (from 28 to 72 hpf) in egg water containing 0.1% (w/v) PTU. After treatment, embryos and larvae were washed twice with egg water containing 0.1% (w/v) PTU, anesthetized with 0.2% (w/v) tricaine for 5 min and fixed in 4% PFA at room temperature for 2 hr. The CLICK-IT reaction for EdU labeling was performed as per manufacturer's protocol (Invitrogen). Samples were processed for immunostaining with anti-GFP, anti-mCherry, and DAPI using the procedure described below.

## Immunostaining

Zebrafish embryos and larvae were collected at different stages and fixed in 4% FPA at room temperature for 2 hr. Next, animals were washed with PBS/1% BSA/1%DMSO/0.5% Triton X-100 (PBDT) and blocked with PBDT/10% goat serum for 1 hr before incubating in primary antibody at 4°C overnight. Samples were washed in PBDT for 30 min × four times and incubated in secondary antibody for 2 hr at room temperature and then incubated with 1 µg/ml DAPI for 10 min and washed with PBS/0.1% Tween.

Primary antibodies used were GFP and mCherry. Phalloidin Alexa Fluor 568 was used to mark F-actin. Secondary antibodies were goat anti-chicken Alexa Fluor 488, goat anti-mouse Alexa Fluor 568, and goat anti-rabbit Alexa Fluor 647.

## Image processing

Confocal data were processed on Imaris x64. Images were prepared using Adobe Photoshop.

## Quantification and statistical analysis

The number of endocardial protrusions at 24 hpf was quantified from maximum intensity projections of the ventricle. The number of endocardial protrusions at 48 hpf was quantified from the mid-sagittal plane of the ventricle. The ratio of trabecular CMs was quantified as previously described (*Jiménez-Amilburu et al., 2016*). The number of trabeculae in the outer curvature of the ventricle was quantified from the mid-sagittal plane, and delaminating CMs were also counted as trabeculae. The number of trabecular CMs in the outer curvature of the ventricle was quantified from the mid-sagittal plane. The number of multilayered CMs in the outer curvature of the ventricle was also quantified from the mid-sagittal plane. EdU$^+$ and Gmnn$^+$ CMs were quantified in the entire ventricle. To quantify the volume of the CJ, the drawing tool paintbrush in Fiji was used to mark the space between the endocardium and myocardium in the outer curvature of the ventricle, followed by a 3D surface reconstruction by Imaris. Systolic and diastolic ventricular areas were measured to calculate ejection fractions. Sample sizes are indicated in each figure, one dot representing one sample. All quantifications were analyzed by the Student's *t*-test (two tailed) and considered significant at p < 0.05. p-Values are indicated in the figures.

## Acknowledgements

We thank Gisela Thana Hartmann, Sarah Howard, Dr Radhan Ramadass, and all fish facility staff for their technical support; Dr Thomas Juan, Dr Samuel Capon, Dr Jordan Welker, Dr Giulia Boezio, and Yiu Chun Law for critical comments on the manuscript; Dr Stefan Baumeister for the schematic model; and Dr Gonzalo del Monte-Nieto for discussion. Research in the Stainier laboratory is supported in part by the Max Planck Society, the DFG (SFB 834/4) and the Leducq Foundation. Research in the Helker laboratory is supported in part by the DFG (SFB 834/4).

# Additional information

## Competing interests
Didier YR Stainier: Senior editor, eLife. The other authors declare that no competing interests exist.

## Funding

| Funder | Grant reference number | Author |
|---|---|---|
| Max-Planck-Gesellschaft | | Didier YR Stainier |
| Deutsche Forschungsgemeinschaft | SFB834 | Didier YR Stainier Christian SM Helker |

The funders had no role in study design, data collection and interpretation, or the decision to submit the work for publication.

## Author contributions
Jialing Qi, Conceptualization, Data curation, Investigation, Methodology, Project administration, Validation, Visualization, Writing – original draft; Annegret Rittershaus, Rashmi Priya, Shivani Mansingh, Resources; Didier YR Stainier, Conceptualization, Funding acquisition, Resources, Software, Supervision, Writing – review and editing; Christian SM Helker, Conceptualization, Funding acquisition, Project administration, Resources, Supervision, Writing – original draft, Writing – review and editing

## Author ORCIDs
Rashmi Priya ⬤ http://orcid.org/0000-0002-0510-7515
Didier YR Stainier ⬤ http://orcid.org/0000-0002-0382-0026
Christian SM Helker ⬤ http://orcid.org/0000-0003-0427-5338

## Ethics
All zebrafish husbandry was performed under standard conditions in accordance with institutional (MPG) and national (German) ethical and animal welfare regulations. All experiments conducted on animals conform to the guidelines from Directive 2010/63/EU of the European Parliament on the protection of animals used for scientific purposes and were approved by the Animal Protection Committee (Tierschutzkommission) of the Regierungspräsidium Darmstadt (Proposal numbers: B2/1017, B2/1041, B2/1138, B2/1218).

## Decision letter and Author response
Decision letter https://doi.org/10.7554/eLife.73231.sa1
Author response https://doi.org/10.7554/eLife.73231.sa2

# Additional files

## Supplementary files
• Transparent reporting form

## Data availability
Figure 2 - Source Data 1, Figure 4 - Source Data 1, Figure 5 - Source Data 1, Figure 6 - Source Data 1, and Supplementary File 1 contain the numerical data used to generate the figures.

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
