## [Editor Report]

This first formal dissection of endocardial protrusions in zebrafish hearts describes how they anchor to cardiomyocytes, and how they participate in signaling pathways involved in trabeculation. The work combines elegant zebrafish reporters and high-quality imaging, as well as mutant lines and pathway inhibitors to provide key findings of how mutual regulation between the myocardium and the endocardium contribute to understanding of mechanisms underlying organ development. This manuscript is of broad interest to readers who study cardiogenesis and developmental biology.

---

## [Decision Letter]

**Decision letter after peer review:**

Thank you for submitting your article "Apelin signaling dependent endocardial protrusions promote cardiac trabeculation in zebrafish" for consideration by *eLife*. Your article has been reviewed by 2 peer reviewers, and the evaluation has been overseen by a Reviewing Editor and Edward Morrisey as the Senior Editor. The following individual involved in review of your submission has agreed to reveal their identity: Naoki Mochizuki (Reviewer #1).

Essential revisions:

Note: The recommendations for the authors from reviewers are also provided in full, as several specific corrections and clarifications are requested. Here we summarize the "essential revisions" to be addressed in a revised manuscript:

1) Provide numbers/additional numbers to buttress some of the data (for example: Nrg-OE, quantitative data that protrusions form at the outer curvature of the looped heart to align with chamber specification and trabeculation).

2) Describe exactly how the graphs were assembled, especially the graphs in Figure 5E-G; they appear to have been extensively manipulated manually.

3) Define terms and stages better: distinguish multi-layering from trabeculation clearly and use the terms accordingly; clearly define the steps of trabeculation early on and then interpret the data in light of the step(s) affected.

4) Modify conclusions where data does not strongly support the conclusions, especially in terms of spatial concordance between protrusions and extruding CMs, and modify the Discussion appropriately.

5) Better description of the mutants used and their overall phenotype.

6) Better discussion of Notch which is dealt with superficially, and better discussion of differences between mouse and fish trabeculation.

*Reviewer #1 (Recommendations for the authors):*

Strengths:

The authors used the transgenic zebrafish lines to show clear images of protrusions from the EdCs and spatiotemporally analyzed them in the transgenic lines: Apln-/-. The importance of protrusions from the EdCs was confirmed by endothelial cell-specific overexpression of dominant-negative form of IRSp53. In addition, to clarify Apln-dependent signaling, the localization of Aplnrb was monitored by the TgBAC(aplnrb:VenusPEST) embryos. Erk activation that might be required for trabeculation was demonstrated in the Tg (myl7:ERK-KTR-Clover-p2a-H2B-tagBFP/mScarlet) larvae. Clear in vivo images followed by statistical analyses support the authors claims well in this paper.

Potential improvements:

1. One might feel difficulty to understand why Nrg2 overexpressed in the endocardium is not enough to activate ErbB on the myocardium and require protrusions that forms anchor points. What do protrusion do for the myocardium? This should be discussed for helping the readers, although the requirement of protrusions is obviously proved by the data. Apln released from the myocardium layer can reach the endocardial layer through cardiac jelly, suggesting that Nrg2a can also reach the myocardium. One thing that might puzzle the readers is how the protrusions modulate Nrg2/ErbB signaling by touching to cardiomyocytes.

2. Are the cardiomyocytes of outer curvature responsive to Apln? According to the data shown in Figure 2F and 2F', distribution of protrusion is dominant in the outer curvature over inner curvature (48 hpf). However, the expression of aplnrb is clearly observed in the inner curvature (Figure 3D', 72 hpf). Therefore, considering the authors' hypothesis, the presence of protrusions might vary during embryogenesis. The protrusion at 72 hpf should be counted as Figure 2F and 2F' to more convincingly demonstrate the relevance of Apln-Aplnrb signaling to protrusions.

Please reconsider the discussion (line 281-288).

3. Similar inconsistency is found in Figure 4I. The number of arrows at region i is more that region ii or region iii in Figure 4C'. This should be reanalyzed to show exact number of protrusions at 48 hpf and 72 hpf.

4. In relation to 1, how are the contacts (touchdowns) kept between protrusions and cardiomyocytes? Are cell-cell adhesion molecules involved in this attachment? How are cytonemes maintained during attachment? This might be interesting to readers because the protrusions are very interesting findings and never detached during contraction.

*Reviewer #2 (Recommendations for the authors):*

1. I do not think that the authors show quantitative data that protrusions form at the outer curvature of the looped heart to align with chamber specification and trabeculation. I think it would be helpful to show this.

2. Page 3. I think that the claim that endocardial protrusions can be detected in "close proximity" to trabecular CMs is not evident from the data. There is perhaps an issue here that might confuse readers – that is, how trabeculae versus multilayering is defined. I understand from previous literature that the first EMT-like events that lead to extrusion of CMs from the outer layer are considered the first step of trabeculation in the fish heart. Trabecular is more traditionally defined as the formation of CM clusters that assemble and protrude towards the lumen. How then should we distinguish between trabeculation as above and multilayering. I think this is unclear in this manuscript and the terms such as "# of trabecular CMs", "# of trabeculae" and "# of multilayered CMs", as used in the various figures, are not properly defined in the paper. Getting back to the point on page 3, the segmentation and rendering of surfaces in video 2 does not allow clear visualization of individual CMs and I do not think there is sufficient evidence that protrusions track more specifically along the side of delaminating or trabecular CMs. This would need further support.

3. Page 3 bottom: I am not sure that 54% of EdU+ CMs being in close proximity to endocardial protrusions can be interpreted in any other way than random distribution of protrusions among CMs (dividing or otherwise). Please modify.

4. Given the points made above, I am unclear by the first sentence of paragraph 2 on page 5 stating "Since we observed a correlation between endocardial protrusions and myocardial trabeculation…"

5. Page 5: experiments to overexpress Irsp53dn in endocardial cells. Experiments were performed at 48 hrs when the interaction between protrusions is complex. If possible, it would be helpful to see data at 24 hrs before endocardial:CM interactions as in Figure 1A and as for Apln mutants in Figure 4.

6. The expression of aplnrb in wholemount embryos shown in Figure 3- figure suppl 1 apparently show higher expression in specific regions at 48hrs. This seems at odds with the expression pattern shown in Figure 3 at 48 hrs. Are the wholemounts useful? Aplnra also shows focal expression in the heart region yet is claimed not to be expressed in heart. For clarity, these sites could be identified?

7. Figure 4 -suppl 1C is not particularly useful since the heart is hardly visible.

8. Page 9: analysis of Aplnrb mutants. More details are needed here about these mutants and whether the general demise of the majority mutants might relate to the cardiac phenotype in the few survivors. Might embryos be delayed, for example? What is the impact of mutation on other vascular beds and how might that impact the cardiac phenotype? Same question applies to Apln mutations and inhibitor treatments. These are global not conditional mutants.

9. The notch reporter does not seem to detect notch expression in CMs as previous reported, which plays a key role in selection of trabeculae and the range of neuregulin signaling.

10. Page 11 and Figure 5 relating to nrg2 over-expression. The phenotype seems marginal in the figure panels B, B' and this is complicated by the lack of clarity around definitions of trabeculation, multilayering etc as noted above. The statistics in Figure 5E for # of trabeculae are marginal and are driven by 3 outliers. Even if there were more multicellular trabecular protrusions, multilayering is proceeding just fine. Is multilayering in over-expressors distinct from the normal trabeculation process? Some clarity needed here. High sample numbers may be needed. Likewise, in nrg2 overexpressing apln mutants, multilayering is advanced, unlike in irsp53 over-expressors. This important section is overall rather weak and the conclusions need better justification. The text refers to Figure 5C-C' and Figure 5D-D' – there are no C', C', D' or D' panels.

11. Discussion. There appear to be key differences between mouse and zebrafish. In particular, endocardial sprouts and touchdowns in mice are cellular, whereas protrusions in zebrafish appear to akin to filopodia. Furthermore, in the early looping heart, the myocardium is multi-layered as sprouting begins. Thus on page 15 it is technically incorrect to say that "In mouse, endocardial sprouting and touchdown formation occur early during cardiac trabeculation.." if we are to consider generally that multilayering is part of trabeculation. I think it would help the reader to be aware of these differences and some considered comments made might be helpful.

12. Likewise, the complexities of notch signaling in zebrafish are relevant and not addressed. If notch function in CMs is important in selecting individual CMs for extrusion from the outer layer, how might endocardial protrusions interact with this process?

13. Schematic model: I think the idea that protrusions have a preferred interaction with extruding CMs does not have strong enough experimental support and it is premature for the figure to reflect that.

14. In graphical figures, such as those in Figure 5 E-G, the graphs appear to have been assembled manually. When blown up, the building blocks of the graph – e.g single or multiple dots – seem to be visible. There are inconsistencies in the shape of dots, the symmetrical or non-symmetrical arrangement of dots, and the thickness and texture of lines showing means and error bars. The authors should assure the reviewers and journal that these are authentic representations of the data and that the statistics have been done appropriately.

---

## [Author Response]

Essential revisions:1) Provide numbers/additional numbers to buttress some of the data (for example: Nrg-OE, quantitative data that protrusions form at the outer curvature of the looped heart to align with chamber specification and trabeculation).

We thank the reviewers for these valid points.

(1.1) We analyzed more larvae and added the data to Figure 5E.

(1.2) It has been shown that cardiac trabeculation mainly occurs in the outer curvature of the ventricle in zebrafish (Liu et al., 2010; Rasouli and Stainier, 2017). Figures 2F’ and 4I (WT and apln^+/+^) show that endocardial protrusions are mainly located in the outer curvature of the ventricle at 48 hpf. To strengthen the conclusion that there is a spatial correlation between endocardial protrusion formation and myocardial trabeculation, we analyzed endocardial protrusions at the onset of cardiomyocyte (CM) delamination (60 hpf) as well as at 72 hpf. We found that in the ventricle, 79 % (at 60 hpf) and 83 % (at 72 hpf) of all endocardial protrusions were located in the outer curvature (Figure 1—figure supplement 1D-F). 69 % of all endocardial protrusions in the outer curvature were close to delaminating CMs at 60 hpf, and 91 % of all endocardial protrusions in the outer curvature were close to trabecular CMs at 72 hpf (Figure 1figure supplement 1D, E, and G). Moreover, 98% of delaminating CMs and 93% of trabecular CMs were in close proximity to endocardial protrusions at 60 and 72 hpf, respectively (Figure 1—figure supplement 1D, E, and H). We included these additional data in Figure 1—figure supplement 1C-H.

2) Describe exactly how the graphs were assembled, especially the graphs in Figure 5E-G; they appear to have been extensively manipulated manually.

We thank the reviewers for this valid point. The inappropriate quality of the graphs was due to the transfer of the graphs from prism to illustrator. We have now exported all graphs with high resolution and added them to the figures.

3) Define terms and stages better: distinguish multi-layering from trabeculation clearly and use the terms accordingly; clearly define the steps of trabeculation early on and then interpret the data in light of the step(s) affected.

We thank the reviewers for the helpful suggestion. Trabecular CMs form “finger-like” multicellular protrusions that bulge out from the compact myocardial layer, whereas multi-layered CMs do not form “finger-like” protrusions and stay close to the compact layer. This distinction can be seen in the data from Lai et al., 2018 (see Figure 2C in their article). We added a description of trabecular CMs and multi-layered CMs in the main text and a schematic illustration in Figure 5—figure supplement 1A and B.

4) Modify conclusions where data does not strongly support the conclusions, especially in terms of spatial concordance between protrusions and extruding CMs, and modify the Discussion appropriately.

As mentioned above, we analyzed endocardial protrusions at the onset of cardiomyocyte (CM) delamination (60 hpf) as well as at 72 hpf. We found that 69 % of all endocardial protrusions in the outer curvature were close to delaminating CMs at 60 hpf, and 91 % of all endocardial protrusions in the outer curvature were close to trabecular CMs at 72 hpf (Figure 1—figure supplement 1D, E, and G). Moreover, 98% of delaminating CMs and 93% of trabecular CMs were in close proximity to endocardial protrusions at 60 and 72 hpf, respectively (Figure 1—figure supplement 1D, E, and H).

We included these additional data in Figure 1—figure supplement 1D-H.

5) Better description of the mutants used and their overall phenotype.

We have further examined the morphology of the heart, the heart rate, and the blood circulation in apln mutants compared with their wild-type siblings, and could not detect significant differences. These new data have been included in Figure 4—figure supplement 3C and D, and Figure 4-videos 1-2.

6) Better discussion of Notch which is dealt with superficially, and better discussion of differences between mouse and fish trabeculation.

We thank the reviewers for these valid points.

(1.1) Loss of the Notch receptor in the endothelium leads to endocardial touchdown defects in mouse (Del Monte-Nieto et al., 2018). To investigate whether Notch signaling plays a similar role in zebrafish, we treated zebrafish embryos with a Notch inhibitor and observed a reduction of touchdowns (Figure 4—figure supplement 4C, D and G), in agreement with the mouse data (Del Monte-Nieto et al., 2018). Furthermore, inhibition of Notch signaling resulted in an increased number of endocardial protrusions (Figure 4—figure supplement 4E, F and H). Whether or not inhibition of Notch signaling also leads to more endocardial protrusions in mouse has not been reported, to the best of our knowledge. These new data can be found in Figure 4—figure supplement 4C-D and G.

(1.2) Endocardial sprouts were first reported by Monte-Nieto et al. in mouse (2018). Endocardial sprouts in mouse appear to be cellular (Monte-Nieto et al., 2018; Figure 1A), whereas endocardial protrusions in zebrafish appear more similar to filopodia (Figure 1), although these differences may be due to the fact that fixed tissue was used for the mouse work compared with live tissue for the zebrafish work. In mouse, the myocardium is already multi-layered before endocardial sprouting begins, whereas in zebrafish the myocardium is single-layered when endocardial protrusions appear. These differences between mouse and zebrafish have been added to the discussion.

References:

Scott, I. C., Masri, B., D'amico, L. A., Jin, S. W., Jungblut, B., Wehman, A. M., Baier, H., Audigier, Y. and Stainier, D. Y. 2007. The g protein-coupled receptor agtrl1b regulates early development of myocardial progenitors. Dev Cell, 12, 403-13.

Zeng, X. X., Wilm, T. P., Sepich, D. S. and Solnica-Krezel, L. 2007. Apelin and its receptor control heart field formation during zebrafish gastrulation. Dev Cell, 12, 391-402.

Liu, J., Bressan, M., Hassel, D., Huisken, J., Staudt, D., Kikuchi, K., Poss, K. D., Mikawa, T. and Stainier, D. Y. 2010. A dual role for ErbB2 signaling in cardiac trabeculation. Development, 137, 3867-75.

Han, P., Bloomekatz, J., Ren, J., Zhang, R., Grinstein, J. D., Zhao, L., Burns, C. G., Burns, C. E., Anderson, R. M. and Chi, N. C. 2016. Coordinating cardiomyocyte interactions to direct ventricular chamber morphogenesis. Nature, 534, 700-4.

Rasouli, S. J. and Stainier, D. Y. R. 2017. Regulation of cardiomyocyte behavior in zebrafish trabeculation by Neuregulin 2a signaling. Nat Commun, 8, 15281.

Lai, J. K. H., Collins, M. M., Uribe, V., Jimenez-Amilburu, V., Gunther, S., Maischein, H. M. and Stainier, D. Y. R. 2018. The Hippo pathway effector Wwtr1 regulates cardiac wall maturation in zebrafish. Development, 145.

Del Monte-Nieto, G., Ramialison, M., Adam, A. A. S., Wu, B., Aharonov, A., D'uva, G., Bourke, L. M., Pitulescu, M. E., Chen, H., De La Pompa, J. L., Shou, W., Adams, R. H., Harten, S. K., Tzahor, E., Zhou, B. and Harvey, R. P. 2018. Control of cardiac jelly dynamics by NOTCH1 and NRG1 defines the building plan for trabeculation. Nature, 557, 439-445.

Helker, C. S., Eberlein, J., Wilhelm, K., Sugino, T., Malchow, J., Schuermann, A.,

Baumeister, S., Kwon, H. B., Maischein, H. M., Potente, M., Herzog, W. and Stainier, D.

Y. 2020. Apelin signaling drives vascular endothelial cells toward a pro-angiogenic state. e*Life*, 9.

Priya, R., Allanki, S., Gentile, A., Mansingh, S., Uribe, V., Maischein, H. M. and Stainier, D. Y. R. 2020. Tension heterogeneity directs form and fate to pattern the myocardial wall. Nature, 588, 130-134.